# Associations of Body Composition, Maximum Strength, Power Characteristics with Sprinting, Jumping, and Intermittent Endurance Performance in Male Intercollegiate Soccer Players

**DOI:** 10.3390/jfmk6010007

**Published:** 2021-01-07

**Authors:** Ai Ishida, S. Kyle Travis, Michael H. Stone

**Affiliations:** Center of Excellence for Sport Science and Coach Education, Department of Sport, Exercise, Recreation, and Kinesiology, East Tennessee State University, Johnson, TN 37604, USA; TRAVISSK@mail.etsu.edu (S.K.T.); STONEM@mail.etsu.edu (M.H.S.)

**Keywords:** collegiate athletes, athlete monitoring, body mass, speed

## Abstract

The purpose of this study was to determine the relationships between body composition, strength, power characteristics, sprinting, jumping, and intermittent endurance performance in collegiate male players. Twenty-three players participated (19.7 ± 1.6 yrs; 71.8 ± 7.1 kg; 176.5 ± 5.1 cm). Measurements of interest in body composition included body fat percentage (BF%), lean body mass (LBM), and body mass (BM). Power characteristics were measured with an unloaded squat jump (SJ0) and loaded SJ at 20 kg (SJ20) and 40 kg (SJ40), and unloaded countermovement jump (CMJ0). Power assessments included peak power (PP) and PP allometrically scaled (PPa). Strength characteristics were assessed using isometric mid-thigh pull. Strength assessment included isometric peak force (IPF) and IPF allometrically scaled (IPFa). Performance measures included 10m and 20 m sprint time, CMJ0 jump-height, and Yo-Yo intermittent endurance test level 1 distance. Significant correlations ranging from moderate to very large were found for LBM and CMJ jump height (CM0 JH) (*p* = 0.01, *r* = 0.50); BF% and sprint times at 10 m (*p* = 0.03, *r* = 0.44) and 20 m (*p* = 0.02, *r* = 0.50). PP and PPa from SJ0 and CMJ0 were significantly correlated to 10m sprint time (*p* < 0.05, *r* = −0.45 to −0.53) and 20 m sprint time (*p* < 0.05, *r* = −0.40 to −0.49). Our findings agree with previous literature in that body composition and power characteristics are directly related to soccer-related performance.

## 1. Introduction

Soccer is one is the most popular sports in the world that can be categorized as an intermittent sport consisting of sprinting, walking, jogging, jumping, kicking, and heading (i.e., soccer-related performance) [1]. Due to the necessary skills and physical demands, soccer players need to have a multifaceted physical capacity, including anaerobic and aerobic aspects needed to repeatedly sprint and jump during match-play. According to previous literature, maximum strength and power are crucial factors to improve the multifaceted capacity, such as sprinting and jumping [2,3], and endurance performance [4,5]. For example, very large to nearly perfect correlations were observed between a half squat and vertical jump height (*p* = 0.02, *r* = 0.78) and 10m sprint time (*p* < 0.001, *r* = 0.94), respectively in elite male soccer players [6]. Muscle strength and power output also have been directly associated with a player′s ability to perform sprints (e.g., getting away from a defender) and jumps (e.g., attempting to head a ball) in male soccer [7,8,9]. Therefore, it may be beneficial for soccer players to improve maximum strength and power, which could lead to maximizing soccer-related sports performance.

Body composition is another important fitness characteristics in soccer [10,11,12,13,14]. Previous literature [10,11,12,14] indicates that lean body mass (LBM) and body fat percentage (BF%) are related to sprinting and jumping performance in male soccer players. Based on previous literature [13,15,16], competitive senior male soccer players maintain LBM of 63.3 to 71.2 kg, with BF% ranging from 10% to 12%. If players have a high LBM (>63.3 kg) with a low BF% (<12%), their sprinting, jumping, and aerobic capacity are typically superior [17,18]. However, inverse relationships have been found between body mass (BM) and jumping and sprinting performance in soccer players [11,12,14]. Radzimiński et al. [14] reported that a large inverse relationship was observed between the percentage fat mass and match-sprinting velocity (*p* < 0.001, *r* = −0.57). These data indicated that when increasing BM, careful monitoring of LBM and BF% and performance should take place as BM gain may not be beneficial in soccer players.

Maximum strength and power characteristics, along with body composition, are primary factors affecting sprinting, jumping, and endurance capabilities in soccer [1,3,6,11,12,14]. However, no data are available for the relationships between body composition, strength, and power characteristics and sprinting, jumping, and/or endurance abilities. Additionally, there are no consistent findings as to the effects of body composition related to physical performance in soccer players [11,12,14]. Understanding the relationships between body composition, strength, and power characteristics relative to soccer-related performance may provide a more comprehensive overview of what characteristics should be considered for assessing performance capabilities. Therefore, the purpose of this study was to determine the relationships between body composition, strength and power characteristics, and sprinting, jumping, and intermittent endurance performance in collegiate male soccer players. Based on the literature and anecdotal observation, we hypothesize that (a) body composition will be strongly correlated to jumping and sprinting performance, (b) body composition will be moderately correlated to endurance performance, and (c) a substantial relationship will be observed between strength characteristics and jump and sprint variables.

## 2. Materials and Methods

### 2.1. Subjects

Twenty-three male soccer players of a single National Association of Intercollegiate Athletics soccer team participated in this study (age: 19.7 ± 1.6 yrs; weight: 71.8 ± 7.1 kg; height: 176.5 ± 5.1 cm; weight training experience: 1 to 4 yrs). All players signed an informed consent document prior to participation, and the study was approved by the University′s Institutional Review Board. The data collection was performed as part of an on-going athlete monitoring program.

### 2.2. Procedure

The testing protocol for body composition, strength, and power assessments was implemented in conjunction with previously reported protocols from our laboratory [19]. All players were familiarized with the testing protocols. Body composition assessments were performed using skinfolds measured by an International Society for the Advancement of Kinanthropometry (ISAK) certified investigator. Prior to the assessment, hydration status was assessed using a refractometer (ATAGO, Tokyo, Japan). If urine specific gravity (USG) was <1.020, players were considered hydrated. If the USG was ≥1.020, athletes drank water until USG indicated a hydrated state. Body composition measurements of interest included BF%, LBM, and BM. A test-retest reliability of skinfold measurement showed a low Coefficient of Variation (CV) of 5.3% [20]. In our laboratory, using an ISAK certified tester, intra-tester reliability for skinfold measures have consistently shown excellent intraclass correlation coefficients (ICC) of ≥0.90 and CVs ≤ 6.0%.

Power characteristics were assessed in a lab setting using unloaded squat jump (SJ0), loaded SJ at 20 kg (SJ20) and 40 kg (SJ40), and unloaded countermovement jump (CMJ0). Players performed a standardized warm-up consisting of 25 jumping jacks, 1 × 5 reps with 20 kg, and 3 × 5 with 60 kg in mid-thigh pull (MTP). Squat jump (SJ) warm-ups included two trials of SJ0, SJ20, SJ40 at 50% and 75% of their perceived maximum efforts, respectively. Countermovement jump (CMJ) warm-up included one trial of CMJ0 at 75% of their perceived maximum efforts. Each load was placed on their shoulders, similar to bar placement used during a back squat. In SJ testing, the players were instructed to stand still on dual force plates (91.0 cm × 91.0 cm; Rice Lake Weighing Systems, Rice Lake, WI, USA) and maintain a squat position at a 90° knee angle measured with a goniometer. The athletes vertically jumped from the squat position with a minimal countermovement on the command of “3, 2, 1, Jump!”. If a tester visually observed evidence of a countermovement from the force-time curve (>200 N), the trial was eliminated and the trial repeated. CMJ testing was performed after SJ testing. Players stood still on the dual force plates and then vertically jump after flexing the hip, knee, and ankle joints on the command of “3, 2, 1, jump”. If the two trials differed by more than 2.0 cm in jump height, another trial was ‘′repeated in SJ and CMJ. The mean of the best two trials was used for data analysis for each condition. CMJ jump height (CMJ JH) was included as a variable relative to soccer-related performance, as players often perform CMJs during match play. Power assessments were determined by peak power (PP; W) and PP allometrically scaled (PPa; W∙kg) from SJ0, SJ20, SJ40, and CMJ. The study showed moderate to excellent test-retest reliabilities in SJ and CMJ JH (SJ0, ICC = 0.98; SJ20, ICC = 0.98; SJ40, ICC = 0.98; CMJ, ICC = 0.99), PP (SJ0, ICC = 0.89; SJ20, ICC = 0.95; SJ40, ICC = 0.90; CMJ, ICC = 0.95), and PPa (SJ0, ICC = 0.79; SJ20, ICC = 0.92; SJ40, ICC = 0.84; CMJ, ICC = 0.94). The kinetic variables also showed low to acceptable CVs for JH (SJ0, CV = 3.7%; SJ20, CV = 3.4%; SJ40, CV = 4.5%; CMJ, CV = 2.7%), PP (SJ0, CV = 7.0%; SJ20, CV = 4.3%; SJ40, CV = 6.2%; CMJ, CV = 1.6%), and PPa (SJ0, CV = 8.0%; SJ20, CV = 4.8%; SJ40, CV = 7.1%; CMJ, CV = 6.7%).

After the CMJ test, isometric mid-thigh pull (IMTP) testing was performed on dual force plates (Rice Lake Systems, Rice Lake, WI, USA; 1000 Hz sampling rate). Players were instructed to flex knee joints to 125 ± 5° measured by a goniometer and to maintain an upright torso with extended elbows. An ITMP warm-up consists of two submaximal trials at 50% and 75% of their perceived maximal efforts. The players pulled upward as fast and hard as possible on the commands of “3, 2, 1, Pull!”. If a tester observed a countermovement in the trial (<200 N), the trial was eliminated and repeated. The average of the best two trials in isometric peak force (IPF) was assessed for data analysis. Variables of interest relating to strength were IPF (N) and IPF allometrically scaled (IPFa; N∙kg). The CVs in IPF and IPFa were low in this study (IPF: CV = 3.6%; IPFa: CV = 3.7%). The test-retest reliabilities for IPF and were excellent (ICC = 0.96 and 0.94, respectively).

A 20 m sprint test was performed on artificial grass using electronic wireless dual eye timing gates (Timing Ireland, Malahide, Dublin, Ireland). The sprint time was split at 10 m and 20 m. The starting timing gates were set approximately at knee height while the height for 10 m and 20 m were at a hip height. Players stood with a staggered stance 30 cm behind the start line and began sprinting of their own volition. The players completed a standardized warm-up and two sprints at 50% and 75% of their perceived maximum efforts before completing at least two sprints at maximal efforts. The best sprint times (s) at 10 m and 20 m were used for data analysis. The CVs between the two trials were 4.0% and 3.1% for 10 m and 20 m sprints, respectively. The test-retest reliabilities in 10 m and 20 m sprint times were poor and moderate (ICC = 0.42 and 0.65, respectively).

A Yo-Yo intermittent endurance test level 1 (YYIR1) was performed to assess soccer-related sprint performance after the 20 m sprint test. The YYIR1 test consists of a 20 m shuttle run at incremental velocity increases with a 10 m active recovery. The initial velocity began at 10 km·h^−1^. Beeps at timed intervals from an audio disk dictated running average velocity. A test was deemed incomplete if a player did not reach the starting line before a beep or if a player was unable to continue. The total distance covered from the YYIR1 was used for data analysis. A systematic review study by Grgic et al. [21] in youth soccer players showed good test-retest reliability (ICC ≥ 0.90) with low CV (<10%).

### 2.3. Statistical Analyses

Pearson correlation coefficient tests were performed to identify the relationships between body composition, strength, and power characteristics and sprinting, jumping, and intermittent endurance performance using RStudio (version 1.1.463, Boston, MA, USA). Correlations were determined using Hopkins classifications [22]: trivial (0–0.09), small (0.1–0.29), moderate (0.3–0.49), large (0.5–0.69), very large (0.7–0.89), nearly perfect (0.9–0.99), and perfect (1.0). Statistical significance was set at *p* ≤ 0.05. All data were expressed as means ± standard deviations.

## 3. Results

Statistically significant large correlations were found between LBM and CMJ JH (*p* = 0.01, *r* = 0.50). Statistically significant moderate to large correlations were observed between BF% and sprint times at 10 m (*p* = 0.03, *r* = 0.44) and 20m (*p* = 0.02, *r* = 0.50) (Table 1). However, non-statistically significant correlations were observed for YYIR1 and LBM (*p* > 0.05, *r* = −0.09), BF% (*p* > 0.05, *r* = −0.05), and BM (*p* > 0.05, *r* = −0.12).

Statistically significant correlations ranging from moderate to very large were found for SJ0 PP and 10m sprint (*p* = 0.03, *r* = −0.45), 20m sprint (*p* = 0.05, *r* = −0.40), and CMJ JH (*p* < 0.001, *r* = 0.83). There were also moderate to very strong correlations for SJ0 PPa and 10 m sprint (*p* = 0.01, *r* = −0.53), 20 m sprint (*p* = 0.02, *r* = −0.48), and CMJ JH (*p* < 0.001, *r* = 0.88). In SJ20, moderate to very large correlations were also observed between PP and sprint times at 10 m (*p* = 0.04, *r* = −0.44) and CMJ JH (*p* < 0.001, *r* = 0.84). SJ20 PPa were statistically correlated to 10 m sprint (*p* = 0.01, *r* = −0.51) and 20 m sprint (*p* = 0.02, *r* = −0.47). In CMJ, moderate to large correlations were found between PP and 10 m sprint (*p* = 0.01, *r* = −0.52) and 20 m sprint (*p* = 0.02, *r* = −0.49). However, non-statistically significant small correlations were observed for IPF and 10 m (*p* > 0.05, *r* = −0.17); IPF and 20 m (*p* > 0.05, *r* = −0.18); IPF and YYIR1 (*p* > 0.05, *r* = −0.25) (Table 2).

Large to very large correlations between LBM and SJ0 PP (*p* < 0.001, *r* = 0.72), SJ0 PPa (*p* = 0.006, *r* = 0.55), SJ20 PP (*p* < 0.001, *r* = 0.72), and SJ20 PPa (*p* = 0.003, *r* = 0.59) were found. Moderate to large correlations were also seen between LBM and CMJ PP (*p* < 0.001, *r* = 0.72), and CMJ PPa (*p* = 0.03, *r* = 0.44) (Table 3). Additionally, statistically significant large correlations were observed between BM and SJ0 PP (*p* = 0.001, *r* = 0.62), SJ20 PP (*p* = 0.002, *r* = 0.62), and SJ20 PPa (*p* = 0.04, *r* = 0.42). Statistically significant moderate to large correlations were observed between IPF and SJ0 PP (*p* = 0.01, *r* = 0.52), SJ0 PPa (*p* = 0.049, *r* = 0.41), SJ20 PP (*p* = 0.01, *r* = 0.50), and SJ20 PPa (*p* = 0.046, *r* = 0.42).

## 4. Discussion

The purpose of this study was to examine the relationships between body composition, strength and power characteristics, and sprinting, jumping, and intermittent endurance performance in collegiate male soccer players. Our hypothesis was generally substantiated in that substantial relationships were observed between isometric strength and power performance, and body composition was largely related to sprinting ability. However, we reject our hypothesis in that only trivial relationships were found between body composition and YYIR1 performance. Our primary findings agree with previous literature in that body composition, strength, and power characteristics are strongly related to soccer-related performance [11,12].

In this study, BF% was positively correlated to 10 m and 20 m sprint performance in collegiate soccer players, while the correlation of LBM to the sprint performance times was negative. In soccer, our findings agree with evidence that positive body composition characteristics (relatively low BF% (<12%) and relatively high LBM (>63.3 kg)) may improve short and long sprint performances [11,12,14]. For example, Silvestre et al. [11] reported that positive correlations were observed between BF% and 9.1 m sprint (*p* < 0.001, *r* = 0.60) and 36.5 m sprints (*p* < 0.001, *r* = 0.61). Our findings also showed no statistically significant correlations between BM and soccer-related sports performance. Supporting these findings, Lago-Peñas et al. [23] found that no substantial difference in BM was noted between successful and unsuccessful male soccer players. Furthermore, we observed that higher LBM and BM could increase jump power output. Therefore, based on our findings and previous literature [11,12,14,23], improving LBM and BF% could positively impact sprinting performance, although excess BM may decrease short sprinting performance in soccer players.

Although not statistically significant, the correlations of BF% and LBM to YYIR1 indicated a negative association agreeing with previous findings [10,11]. Two possible explanations for these differences between our study and the previous studies [10,11] could be 1) the use of different endurance tests and 2) positional difference in their samples. To further explain these possible differences, in our study, the YYIR1 test was used to assess intermittent endurance capacity, while Silvestre et al. [11] and Radzimiński et al. [10] used Yo-Yo intermittent endurance test level 2 and 20 m multi-stage fitness test, respectively. Additionally, playing position could affect the correlation between body composition and YYIR1. In this study, goalkeepers were excluded due to different physiological capacity. Compared to outfielders (defender, midfielder, and forward), goalkeepers do not have a similar aerobic power due to match-physical demands [24,25]. A meta-analysis by Slimani et al. [25] reported that lower maximum oxygen consumption was observed in goalkeepers compared to defenders (*p =* 0.004, ES = 1.31), midfielders (*p =* 0.001, ES = 1.37), and forwards (*p* = 0.014, ES = 1.07). Positional differences within outfielders may also affect the relationship between body composition and endurance performance due to different match physical demands [25,26,27]. Therefore, further investigation will be required to examine the relationship between body composition and endurance capacity by different positions.

Power is another important factor in physical performance for soccer [1,6,14]. High power production can provide performance advantages for sprinting and jumping in soccer players [8,9,28,29]. Our study provided additional evidence for substantial relationships among the 10 m sprint, SJ0, SJ20, and CMJ0, PP, and PPa. Interestingly, the findings also indicated that PPa from SJ0 and CMJ had a stronger association with sprint performance than the PP. The difference in the magnitude between PP and PPa may be explained by the effects of BM. Allometric scaling (i.e., PPa and IPF) helps to obviate differences in BM and provides insight on normalizing strength and power [30]. Recent investigations [11,31] also have shown that higher BM is associated with slower sprint performance and lower jump heights, while LBM and BF% can positively affect sprint and jump performance. High LBM should improve power output, leading to increasing physical performance, although the increase in BM with non-functional tissues (i.e., fat) could decrease CMJ and sprint performances. Therefore, PPa may be a better indicator of sprint performance than PP in soccer players.

It should also be noted that strength level plays a role in power production and soccer-related physical characteristics such as sprinting and jumping [6,7,29]. For example, Ishida et al. [29] reported that strength was shown to be positively related to power production and sprint ability. However, care should be taken when improving power production associated with physical performance in soccer players. As sports coaches intuitively notice, increased absolute maximal strength and power production may not necessarily improve soccer physical performance. Our study showed large to very large correlations for SJ, PP, and IPF, LBM, and BM, while moderate to large positive correlations were observed for BF% and sprint times. The results of this study suggest that players could sprint slower with increased jump power production when coupled with excessive BM, particularly if fat mass was also relatively high. While strength and power can be increased without BM gains, care must be taken when gaining body mass so as not to gain non-functional mass (e.g., fat). Therefore, careful monitoring of maximum strength and power production should include body composition measurement in soccer players.

There are two primary limitations of this study. First, match-physical performance (i.e., total distance covered and match maximal running velocity) was not assessed. This information would allow researchers to better understand what factors underlie match physical performance as well as lab-based soccer-related sports performance. Second, in this current study, the sample was collegiate male soccer players. It may not be possible to generalize our data to the players at an elite level (i.e., professional and international levels).

## 5. Conclusions

The results of this study indicate that body composition and power characteristics are substantially related to sprinting and jumping performance in male intercollegiate soccer players. Body composition and power characteristics would be essential components in soccer-related performance characteristics such as sprinting and jumping abilities. Improving LBM and BF% could be beneficial for jumping and sprinting performance in soccer, although excessive BM may negatively affect jump and sprint performance. However, it is possible that low LBM could limit players’ jump performance, sprint, and power output. Thus, an optimal balance between LBM and BF% should be desired to enhance performance. Sports scientists, coaches, and athletes may consider implementing training routines and nutritional strategies that aid in improving body composition during the in-season to, in turn, maintain or improve sprinting performance.

## Figures and Tables

**Table 1 jfmk-06-00007-t001:** Correlations between Body Composition and Sprinting, Jumping, and Intermittent Endurance Performance.

Variables	10 m	20 m	CMJJH	YYIR1
Lean Body Mass	−0.33	−0.27	0.50 *	−0.09
BF%	0.44 *	0.50 *	−0.37	−0.05
BM	−0.29	−0.03	0.23	−0.12

Note. 10 m = 10 m sprint. 20 m = 20 m sprint. CMJ = Countermovement jump. YYIR1 = Yo-Yo Intermittent Recovery Test level 1. BF% = Body fat percentage. BM = Body mass. * = denotes *p* < 0.05.

**Table 2 jfmk-06-00007-t002:** Correlations between Maximum Strength and Power Characteristics and Sprinting, Jumping, and Intermittent Endurance Performance.

Variables	10 m	20 m	CMJ	YYIR1
Isometric Peak Force	−0.17	−0.18	0.37	−0.25
IPFa	−0.17	−0.23	0.33	−0.24
SJ0 Peak Power	−0.45 *	−0.40 *	0.83 ^†^	−0.31
SJ0 PPa	−0.53 *	−0.48 *	0.88 ^†^	−0.30
SJ20 Peak Power	−0.44 *	−0.39	0.84 ^†^	−0.32
SJ20 PPa	−0.51 *	−0.47 *	0.88 ^†^	−0.32
SJ40 Peak Power	−0.39	−0.34	0.77 ^†^	−0.33
SJ40 PPa	−0.43 *	−0.39	0.79 ^†^	−0.30
CMJ Peak Power	−0.46 *	−0.42 *	0.91 ^†^	−0.28
CMJ PPa	−0.52 *	−0.49 *	0.93 ^†^	−0.25

Note. 10 m = 10 m sprint. 20 m = 20 m sprint. CMJ = Countermovement jump. YYIR1 = Yo-Yo Intermittent Recovery Test level 1. IPFa = Allometrically scaled isometric peak force. SJ0 = Squat jump with a polyvinyl chloride pipe. PPa = Allometrically scaled peak power. SJ20 = Squat jump with a 20 kg bar. SJ40 = Squat jump with a bar loaded to 40 kg. * = denotes *p* < 0.05. ^†^ = denotes *p* < 0.001.

**Table 3 jfmk-06-00007-t003:** Correlations between Body Composition, Maximum Strength, and Power Characteristics.

Variables	LBM	BF%	BM	IPF	IPFa
SJ0 Peak Power	0.72 ^†^	−0.06	0.62	0.52 *	0.33
SJ0 PPa	0.55 *	−0.19	0.39	0.41 *	0.32
SJ20 Peak Power	0.72 ^†^	−0.08	0.62 *	0.50 *	0.32
SJ20 PPa	0.59 *	−0.19	0.42 *	0.42 *	0.31
SJ40 Peak Power	0.74 ^†^	−0.04	0.64 *	0.46 *	0.25
SJ40 PPa	0.64 ^†^	−0.12	0.50 *	0.38	0.22
CMJ Peak Power	0.71 ^†^	−0.18	0.54 *	0.41	0.24
CMJ PPa	0.44 *	−0.29	0.23	0.32	0.27

Note. LBM = Lean body mass. BF% = Body fat percentage. BM = Body mass. IPF: Isometric peak force. IPFa: Allometrically scaled isometric peak force. PP: SJ = Squat jump. Peak power. PPa Allometrically scaled peak power. CMJ = Countermovement jump. * = denotes *p* ≤ 0.05. ^†^ = denotes *p* < 0.001.

## Data Availability

The data that support the findings of this study are available from the corresponding author upon reasonable request.

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
