# Peer review of "Associations of Body Composition, Maximum Strength, Power Characteristics with Sprinting, Jumping, and Intermittent Endurance Performance in Male Intercollegiate Soccer Players"

_jfmk, 2021, doi:10.3390/jfmk6010007_

Round 1

Reviewer 1 Report

The manuscript is about the relationship of Body Composition, Maximum strength, and power characteristics with soccer- related Physical Performance.

Abstract is correct and concise

Introduction is appropriate.

Methodology

Is there a familiarity with the tests, since, when making jumps with weight, the technique can be modified and influence the procedure and results.

Results

Lines 154-165; 168-175: It becomes very difficult to follow the results with so many numbers, simplify the results including the most important ones.

Discussion

Line 244: You present the conclusions of the research and you do it again in the section of conclusions. It should only appear in the conclusions section.

What practical applications would your results have for coaches, add it to the document.

What does this research contribute to what already exists?

The text is correctly written.

Author Response

Reviewer 1

The manuscript is about the relationship of Body Composition, Maximum strength, and power characteristics with soccer- related Physical Performance.

Abstract is correct and concise, Introduction is appropriate.

Thank you for your review and comments.

Methodology

Is there a familiarity with the tests, since, when making jumps with weight, the technique can be modified and influence the procedure and results.

All the players have completed the testing twice already as a part of athlete monitoring programs. Therefore, they are familiarized with the testing protocol. I added the sentence in Line 81.

Results

Lines 154-165; 168-175: It becomes very difficult to follow the results with so many numbers, simplify the results including the most important ones.

I split into different sentences to make them readable. Additionally, I removed some correlations because I did not mention in the Discussion.  Please see the lines from 152 to 176

Discussion

Line 244: You present the conclusions of the research and you do it again in the section of conclusions. It should only appear in the conclusions section.

What practical applications would your results have for coaches, add it to the document.

What does this research contribute to what already exists?

I deleted Line 244 and add some practical application information in the conclusion section (Line 251 to 260).

Reviewer 2 Report

This manuscript is a report on an investigation in which the extent to which body composition, strength, and power variables were associated with sprinting, jumping, and endurance performance in a sample of male intercollegiate soccer players. The topic addressed in the manuscript falls within the domain covered by this journal and the investigation was conducted with forethought and care. The manuscript is very well-written. These positive impressions notwithstanding, I have several concerns about the manuscript in its current form:

  1. Although the sprinting, jumping, and endurance tasks examined in the current study are relevant to soccer, they are relevant to many other sports as well. Consequently, the title is somewhat misleading. A more accurate title would be something along the lines of “Associations of Body Composition, Maximum Strength, and Power Characteristics with Sprinting, Jumping, and Endurance Performance in Male Intercollegiate Soccer Players.” References to “soccer-related physical performance” elsewhere in the manuscript (e.g., l. 62, l. 180) should also be replaced with “sprinting, jumping, and endurance performance” (or the like).
  2. Are the participants members of a single team? If so, it is possible that the recruiting preferences and practices of the coaching staff may have contributed to the results obtained. Athletes with certain somatotypes may have been recruited to fulfill particular roles on the team and those roles may have overlapped with certain performance attributes. A broader sample with players from an array of teams should be recommended for future research.
  3. On l. 205-211, the authors demonstrate their recognition that positional heterogeneity in the sample (i.e., inclusion of goalkeepers along with the outfielders) can influence the results. Heterogeneity among the outfielders in the current study could also have affected the results. Toward that end, it might be useful to examine the relationships tested in the current study separately for players in specific positions (e.g., central midfielders, wingers/wide backs, strikers).

Author Response

Reviewer2

This manuscript is a report on an investigation in which the extent to which body composition, strength, and power variables were associated with sprinting, jumping, and endurance performance in a sample of male intercollegiate soccer players. The topic addressed in the manuscript falls within the domain covered by this journal and the investigation was conducted with forethought and care. The manuscript is very well-written. These positive impressions notwithstanding, I have several concerns about the manuscript in its current form:

Thank you for your words and comments. My response is highlighted in red.

  1. Although the sprinting, jumping, and endurance tasks examined in the current study are relevant to soccer, they are relevant to many other sports as well. Consequently, the title is somewhat misleading. A more accurate title would be something along the lines of “Associations of Body Composition, Maximum Strength, and Power Characteristics with Sprinting, Jumping, and Endurance Performance in Male Intercollegiate Soccer Players.” References to “soccer-related physical performance” elsewhere in the manuscript (e.g., l. 62, l. 180) should also be replaced with “sprinting, jumping, and endurance performance” (or the like).

As you requested, we changed the title and the change the phrase “soccer-related physical performance” to sprinting, jumping, and intermittent endurance performance.

  1. Are the participants members of a single team? If so, it is possible that the recruiting preferences and practices of the coaching staff may have contributed to the results obtained. Athletes with certain somatotypes may have been recruited to fulfill particular roles on the team and those roles may have overlapped with certain performance attributes. A broader sample with players from an array of teams should be recommended for future research.

In Line 72, I added the line of single National Association of Intercollegiate Athletes soccer team. To provide the players characteristics, I added year’s experience of weight training in this group in Line 74.  

  1. On l. 205-211, the authors demonstrate their recognition that positional heterogeneity in the sample (i.e., inclusion of goalkeepers along with the outfielders) can influence the results. Heterogeneity among the outfielders in the current study could also have affected the results. Toward that end, it might be useful to examine the relationships tested in the current study separately for players in specific positions (e.g., central midfielders, wingers/wide backs, strikers).

I revised the sentences to “positional difference”. And then, as you said, we described the potential effects of position on the relationships.  Please see the lines from 206 to 216.

Round 2

Reviewer 2 Report

I appreciate the efforts of the authors in addressing my concerns and those of the other reviewer. The manuscript has be improved as a result of the changes that have been made.